# SLED: Self-Supervised Dataset Distillation for Lightweight Experience Replay

## Abstract

Experience Replay (ER) is central to off-policy reinforcement learning, but its reliance on massive buffers creates prohibitive storage and sampling costs. We introduce SLED, a self-supervised dataset distillation framework that replaces conventional replay with a compact, learnable synthetic dataset. SLED progressively shifts the agent's training from real interactions to a small, self-evolving knowledge base by decoupling data writing from training sampling. On the writing side, a temporal schedule gradually substitutes real trajectories with optimized synthetic samples, leaving the buffer composed solely of distilled data. On the sampling side, a quota-based strategy shapes the training distribution, enabling a seamless transition from "real-dominated" to "synthetic-dominated" updates without altering the base algorithm. To preserve the long-term utility of synthetic data, SLED adopts an online-validated evolutionary optimization scheme: candidate synthetic datasets undergo brief parallel training trials, followed by real-environment evaluation, yielding a dataset-level fitness signal that guides their continual refinement. The framework is plug-and-play with mainstream off-policy methods. Overall, SLED systematically extends the idea of dataset distillation to the non-stationary regime of reinforcement learning, providing a practical alternative to large-scale replay buffers. Extensive experiments on DMControl, Habitat, and Atari confirm that SLED delivers superior efficiency and scalability over existing ER approaches, demonstrating its broad effectiveness across diverse domains. Our code is available at https://anonymous.4open.science/r/Sled-25F2/.

## 1 Introduction

Experience replay (ER) is a foundational component of modern reinforcement learning (RL) Andrychowicz et al. (2020); Levine et al. (2016); Todorov et al. (2012); Jiang et al. (2024); Zhao & Tresp (2018); Fang et al. (2019), significantly enhancing sample efficiency by allowing agents to reuse past experiences and decorrelate consecutive updates Schaul et al. (2015). In off-policy algorithms such as Soft Actor-Critic (SAC) Haarnoja et al. (2018), ER is implemented through a large replay buffer, which typically stores millions of transitions. These transitions are drawn uniformly or based on heuristic priorities to compute updates for the policy and critic. The scalability and flexibility of such buffers have made ER a standard technique in sparse reward signals Hare (2019); Silver et al. (2017), high-dimensional state spaces Andrychowicz et al. (2020); Levine et al. (2016); Ibrahimi et al. (2012); Wang et al. (2024a), and low sample efficiency Yarats et al. (2021); Wang et al. (2024b); Schrittwieser et al. (2020) environments.

Despite its widespread adoption, conventional ER suffers from several fundamental limitations that hinder its applicability in real-world systems Dulac-Arnold et al. (2019). First, storing and sampling from massive buffers incurs substantial memory and latency costs, which are prohibitive for resource-constrained platforms such as mobile devices or embedded controllers. Second, uniform sampling often leads to inefficient training Kompella et al. (2022), since most transitions contribute little to learning progress. Third, while prioritized experience replay (PER) Schaul et al. (2015) improves over uniform sampling, it relies on static heuristics such as temporal-difference (TD) error. These heuristics are highly sensitive to noise and fail to adapt to non-stationary dynamics or sparse rewards, thereby limiting overall learning efficiency.

To address these challenges, we ask whether a large replay buffer can be replaced by a compact, learnable dataset that preserves its essential learning signal while sharply reducing memory and sampling overhead. Building on advances in dataset distillation Wang et al. (2018); Liu et al. (2022) for supervised learning Lei & Tao (2023), we introduce **SLED**, *Self-supervised Dataset Distillation for Lightweight Experience Replay*, a plug-and-play framework that replaces large replay buffers with a compact, learnable synthetic dataset $D_\phi$ in a standard off-policy loop. The core idea is a **finely-tuned process** of *train–distill–replace*: a unified buffer is progressively shifted from real interactions to synthetic transitions, while the learner's training curriculum is meticulously sculpted to ensure a smooth migration from real-dominated to synthetic-dominated batches. This raises two coupled challenges: (i) migrating dependence from massive real data to a small synthetic set without disrupting learning, and (ii) continually optimizing $D_\phi$ under nonstationarity so it consistently supplies high-value signals.

SLED achieves this through two pillars of fine-grained control. First, a decoupled dual-scheduling mechanism (§3.1) enables precise management of the learning process by separating the buffer's physical contents from the training curriculum. A temporal write schedule $\beta(t)$ orchestrates the gradual replacement of real transitions with synthetic ones, while a separate, quota-based sampling schedule $\alpha(t)$ precisely dictates the real-to-synthetic ratio in each minibatch. This deliberate curriculum management provides an alternative to reactive heuristics like PER, leaving the base learner unchanged. Second, an online-validated evolutionary strategy (§3.2) provides continuous, fine-grained optimization of $D_\phi$'s utility under non-stationarity. Candidate datasets undergo brief parallel training trials and real-environment evaluation to produce a dataset-level fitness signal; mirrored perturbations yield a black-box gradient estimate, and updates are projected onto valid domains to ensure stability. These components integrate seamlessly into standard off-policy algorithms (§3.3), yielding an efficient learning paradigm driven by a compact, self-evolving replay. Beyond design, we provide theoretical analysis showing that SLED bounds and controls a value-weighted discrepancy between real- and synthetic-trained policies, with a return-gap bound scaling linearly in $1/(1-\gamma)$. Empirically, across DMControl, Habitat, and Atari, SLED consistently matches or surpasses Prioritized ER while requiring only a fraction of the storage and incurring negligible wall-clock overhead, demonstrating that large-scale replay can be effectively replaced by compact, self-evolving synthetic datasets. In summary, our contributions are as follows:

- We introduce an online experience distillation paradigm for non-stationary off-policy RL, which elevates a compact, learnable synthetic set to a first-class replay substrate.

- We develop an online validated evolutionary optimization at the dataset level, combining brief parallel training trials and fixed protocol evaluation with mirrored perturbations and validity projections on action, reward, and termination fields to ensure stability under nonstationarity.

- We provide theoretical analysis showing that SLED bounds a value-normalized distributional mismatch between real and distilled training distributions, yielding a return-gap bound that scales linearly with $1/(1-\gamma)$

- We demonstrate plug-and-play effectiveness across DMControl, Habitat, and Atari: SLED matches or surpasses other ER baselines while using only a fraction of the storage, significantly improving sample efficiency with negligible additional wall-clock cost.

## 2 PRELIMINARIES

**Off-Policy Reinforcement Learning.** We consider a standard reinforcement learning setting defined by a Markov decision process (MDP) $(\mathcal{S}, \mathcal{A}, P, r, \gamma)$ with discount factor $\gamma \in [0, 1)$. The agent seeks a policy $\pi_\theta(a \mid s)$ that maximizes the expected discounted return:

$$J(\pi_\theta) = \mathbb{E}_{\pi_\theta, \, P} \Big[ \sum_{t=0}^{\infty} \gamma^t \, r(s_t, a_t) \Big]. \tag{1}$$

In the off-policy regime, we approximate the action-value function $Q^\pi(s,a)$ with a critic $Q_\psi(s,a)$ and train it jointly with the policy (actor) $\pi_\theta$. Interaction with the environment produces transition tuples $x = (s, a, r, s', d)$, where $d \in \{0, 1\}$ denotes termination.

**Experience Replay and its Limitations.** Modern off-policy algorithms (e.g., SAC, TD3) rely on an experience replay buffer $\mathcal{B}_{\text{real}}$ to store past transitions, enabling sample reuse and decorrelated updates. While large buffers (e.g., $N! \sim !10^6$) improve stability, they also introduce substantial memory and sampling overhead, which is particularly problematic in real-time or resource-constrained settings. Moreover, uniform sampling is inefficient, as many transitions contribute little to learning progress. Prioritized experience replay (PER) alleviates this by emphasizing high TD-error samples, but it depends on fixed heuristics that are noise-sensitive and often brittle in non-stationary or sparse-reward environments.

**Synthetic Dataset Distillation.** To address these issues, we introduce a compact, learnable synthetic dataset $D_\phi = \{(s_j, a_j, r_j, s'_j, d_j)\}_{j=1}^m$ with size $m \ll N$. Each transition in $D_\phi$ is parameterized by $\phi$ and iteratively optimized using online learning signals, while being constrained to valid state, action, and reward domains. During training, both the actor and critic are updated on mini-batches that mix real samples from $\mathcal{B}_{\text{real}}$ with synthetic samples from $D_\phi$. This allows $D_\phi$ to serve as a high-value, compact "condensed memory." Optimization of $D_\phi$ is guided by an *online evaluation* mechanism that directly maximizes the expected return of a temporarily trained policy when evaluated on the real task; details are provided in the subsequent methodology section.

# 3 METHODOLOGY

We propose SLED, a plug-and-play framework that replaces large replay buffers with a compact synthetic dataset. Its design rests on three pillars: (i) a decoupled dual-scheduling mechanism that separates storage from the training curriculum (§3.1); (ii) an online-validated evolutionary strategy that maintains the utility of the synthetic dataset under nonstationarity (§3.2); and (iii) seamless integration into standard off-policy algorithms without altering their update rules (§3.3). We next describe each component in detail.

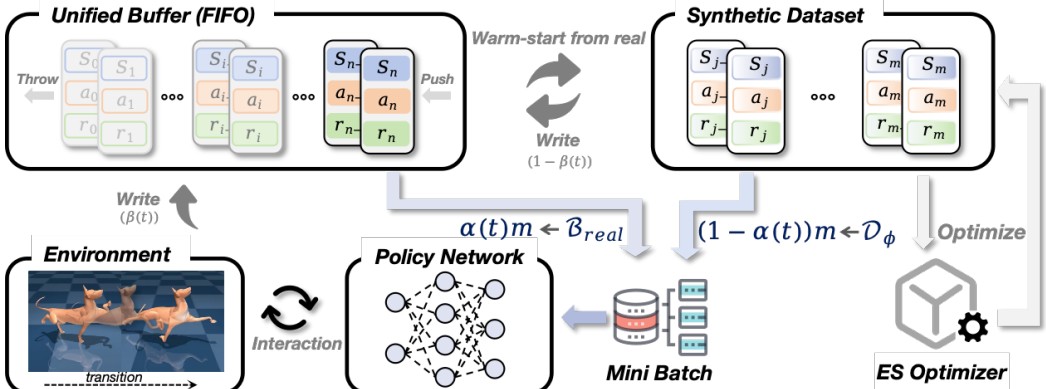

Figure 1: Overview of SLED. The agent interacts with the environment to generate transitions. Writes to the unified FIFO buffer occur as real with probability $\beta(t)$ and as synthetic (from $D_\phi$) with probability $1 - \beta(t)$. During training, a mini-batch of size $m$ is formed by *quota-based stratified sampling*: $m_{\text{real}} = \lfloor \alpha(t)\,m \rfloor$ samples are drawn from the buffer's real sub-population and $m_D = m - m_{\text{real}}$ from $D_\phi$ (backfilled from $D_\phi$ if real is insufficient). These are fed to the policy and critic update, while an ES optimizer asynchronously refines $D_\phi$ via online validation. The key is the full decoupling of the *writing schedule* $\beta(t)$ (storage composition) from the *sampling schedule* $\alpha(t)$ (training curriculum), which prevents short-term buffer fluctuations from contaminating the training distribution and enables a smooth transition from a full replay buffer to a compact distilled dataset.

## 3.1 CORE MECHANISM OF SLED

**Motivation.** Large replay buffers improve sample diversity but incur considerable storage and maintenance costs; replacing real experience entirely with a small synthetic dataset can introduce distributional mismatch and training instability. Our objective, without changing the base algorithm's update rules, is to achieve: (i) memory friendliness with eventual convergence to $D_\phi$-only; (ii) a controllable training distribution with a smooth transition; and (iii) a simple implementation that is easy to reproduce.

**Decoupled Schedules.** Large-scale replay buffers impose heavy storage and maintenance costs, yet directly replacing them with a small synthetic set can trigger abrupt distribution shifts and destabilize learning. Our key insight is to decouple *what is physically stored* in the buffer from *what the learner consumes during training*. When storage and sampling are tied: for example, when the write ratio immediately dictates the sample ratio: any fluctuation in buffer composition propagates directly into the training distribution. This coupling amplifies noise, causing oscillations in the learning signal. By introducing independent schedules for writing and sampling, SLED ensures that buffer churn does not leak into the training process, enabling a smooth and stable shift from real to synthetic experience. We realize this decoupling through two independent schedules, which serve as the primary levers for fine-grained control. A *write schedule* $\beta(t)$ governs the physical composition of the buffer, gradually shifting storage from real to synthetic transitions. In parallel, an independent sampling schedule $\alpha(t)$ shapes the training curriculum by explicitly controlling the real-to-synthetic ratio in each minibatch. Although both schedules are typically monotone (real $\rightarrow$ synthetic), they need not be identical: $\beta(t)$ ensures smooth replacement at the storage level, while $\alpha(t)$ provides direct control over the learner's input distribution. Crucially, minibatch quotas follow $\alpha(t)$ independently of the buffer's current physical composition induced by $\beta(t)$; see Eq. 2–7.

**Writing Schedule $\beta(t)$: Controlling Buffer Composition.** Let $\mathcal{B}_t$ denote the unified buffer at time $t$ with capacity $C$. The buffer operates in a FIFO (First In First Out) manner: one item is written per step; when full, the oldest item is evicted. Each newly written entry at step $t$ comes from either real interaction or $D_\phi$ with probabilities governed by $\beta(t)$:

$$\Pr\big[x \in \mathsf{real}\big] = \beta(t), \qquad \Pr\big[x \in D_\phi\big] = 1 - \beta(t), \qquad \beta(t) \in [0, 1]. \tag{2}$$

Under this FIFO model, the *expected* fraction of real samples in the buffer, $\rho_{\mathrm{real}}(t)$, evolves as

$$\rho_{\mathrm{real}}(t+1) = \Big(1 - \frac{1}{C}\Big)\rho_{\mathrm{real}}(t) + \frac{1}{C}\beta(t). \tag{3}$$

This recursion acts as a low-pass filter on $\beta(t)$, damping short-term shocks in buffer composition. Its closed-form solution, for $\rho_{\mathrm{real}}(0) \in [0, 1]$, makes the smoothing and limit behavior explicit:

$$\rho_{\mathrm{real}}(t) = \Big(1 - \tfrac{1}{C}\Big)^t \rho_{\mathrm{real}}(0) + \tfrac{1}{C}\sum_{k=0}^{t-1}\Big(1 - \tfrac{1}{C}\Big)^k \beta(t-1-k).$$

If more than one item is written per step (e.g., parallel environments with an average of $\kappa$ items per step), replace $1/C$ by $\kappa/C$; the qualitative conclusions below remain unchanged. Eq. 3 is a low-pass recursion on $\beta(t)$, so buffer composition changes smoothly even if $\beta(t)$ momentarily fluctuates.

**Proposition 1** (Asymptotic Property of Buffer Composition). *Assume a unified FIFO buffer of capacity $C$ with one item written per step, initial fraction $\rho_{real}(0) \in [0, 1]$, and a write schedule $\beta(t) \in [0, 1]$ satisfying $\lim_{t\to\infty}\beta(t) = 0$. Let $\rho_{real}(t)$ denote the fraction of real samples in the buffer at time $t$. Then*

$$\lim_{t\to\infty}\mathbb{E}\big[\rho_{real}(t)\big] = 0, \tag{4}$$

Based on Proposition 1, the expected real fraction vanishes asymptotically and the buffer converges to $D_\phi$-only. As $\beta(t) \downarrow 0$, the buffer asymptotically becomes distilled-only storage, enabling memory savings without abruptly cutting off real data early on. See Appendix A.1 for a proof.

**Sampling Schedule $\alpha(t)$: Shaping the Training Curriculum.** Independent of the writing side, $\alpha(t) \in [0, 1]$ directly controls the quotas of real and $D_\phi$ in each training mini-batch. Given a batch size $m$, we apply quota-based stratified sampling:

$$m_{\mathrm{real}}(t) = \big\lfloor m\,\alpha(t)\big\rfloor, \qquad m_D(t) = m - m_{\mathrm{real}}(t). \tag{5}$$

When $m\alpha(t)$ is not an integer, the rounding via $\lfloor\cdot\rfloor$ induces at most a one-sample deviation (stochastic rounding can further reduce bias). If real samples are insufficient, i.e., $N_{\text{real}}(t) < m_{\text{real}}(t)$, the deficit is backfilled by $D_\phi$:

$$m_{\text{real}}^\star(t) = N_{\text{real}}(t), \qquad m_D^\star(t) = m - m_{\text{real}}^\star(t). \tag{6}$$

The target mini-batch training distribution is

$$\pi_t(x) = \alpha(t)\, p_{\text{real}}(x) + \big(1 - \alpha(t)\big)\, p_D(x), \tag{7}$$

where $p_{\text{real}}$ and $p_D$ denote the distributions of real experience and synthetic data, respectively. (Both can vary slowly over time due to buffer/ES updates.) The decoupled design yields two main benefits. **(i) No starvation:** the backfilling rule in Eq. 6 guarantees that the learner always receives sufficient data and can fully exploit all available real samples before transitioning. **(ii) Stability:** the low-pass property of Eq. 3 smooths buffer-side shocks, while Eq. 7 gives precise curriculum control; together they prevent short-term oscillations in the training distribution.

**Proposition 2** (Upper Bound on Curriculum Deviation). *Let $\hat{\pi}_t$ be the empirical training distribution obtained by quota-based stratified sampling (without replacement), and let the target distribution be $\pi_t(x) = \alpha(t)p_{real}(x) + (1-\alpha(t))p_D(x)$. Denote the empirical sub-distributions by $\hat{p}_{real}$ and $\hat{p}_D$. Then the total variation distance between the empirical and target distributions satisfies*

$$\text{TV}\big(\hat{\pi}_t, \pi_t\big) \leq \alpha(t)\,\text{TV}\big(\hat{p}_{real}, p_{real}\big) + \big(1 - \alpha(t)\big)\,\text{TV}\big(\hat{p}_D, p_D\big). \tag{8}$$

This shows that the deviation from the target curriculum is controlled by a convex combination of the approximation errors of the two sub-distributions, weighted by $\alpha(t)$, with $O(1/m)$ fluctuations due to finite batch size. See Appendix A.2 for a proof.

Before running ES, we *warm-start* the synthetic dataset $D_\phi$ with a small set of real transitions. Concretely, we first run a short real-only phase with $\alpha(t) = \beta(t) = 1$ to collect a temporary buffer of recent interactions; then we select $N$ representative transitions and copy them into $D_\phi$ as the initial entries. This yields a valid and informative starting point and avoids purely random noise at $t=0$. Afterwards, $D_\phi$ is refined by the online-validated ES in §3.2 (with projection to feasible domains), while $\beta(t)$ and $\alpha(t)$ are annealed from real-dominated to $D_\phi$-dominated storage and training, respectively.

### 3.2 ES-BASED DATASET OPTIMIZATION WITH ONLINE VALIDATION

Having ensured a smooth transition of the training distribution, we now confront a more fundamental challenge: *how to preserve the long-term utility of the synthetic dataset $D_\phi$ under non-stationary dynamics?* Classical gradient-matching distillation is inadequate in this setting because the ultimate objective: the agent's downstream performance is non-differentiable with respect to $D_\phi$. This objective is only revealed after full training and evaluation, and is further confounded by the current sampling regime and environment stochasticity. Consequently, direct gradients are unavailable and surrogate gradients are unreliable, making gradient-matching unsuitable. This limitation motivates an alternative view: treat $D_\phi$ as a decision variable in a black-box optimization loop that is validated online against real-environment performance, enabling dataset updates that remain aligned with the end task despite non-stationarity.

**Online-validated ES.** Because our end goal is downstream performance that is non-differentiable w.r.t. $D_\phi$, we adopt an evolution-strategies (ES) approach that treats the *entire training-and-evaluation process* as a black-box objective. Instead of relying on gradients, ES iteratively improves $D_\phi$ by measuring the *fitness* of candidate datasets via *brief parallel training trials* run under the current sampling regime, followed by evaluation with a fixed protocol. In this way, the optimization directly targets the ultimate quantity of interest: downstream return.

To formalize the objective, let the learnable synthetic dataset be $D_\phi = \{(s_j, a_j, r_j, s'_j, d_j)\}_{j=1}^N$ with $N \ll |\mathcal{B}_t|$, and let $\phi$ collect all learnable numeric fields. The goal is to maximize downstream return under a fixed evaluation protocol. For any $\phi$, define a *brief parallel training trial* as running $L$ training steps under the current sampling schedule $\alpha(t)$ while using $D_\phi$ as the synthetic source. Let $\mathcal{S}$ be a fixed set of evaluation seeds, and let $\text{TrialReturn}(\phi; s)$ denote the return obtained by evaluating the learner (after the $L$-step trial) under seed $s$. The black-box objective is

$$J(\phi) := \frac{1}{|\mathcal{S}|} \sum_{s \in \mathcal{S}} \text{Return}(\text{TrialReturn}(\phi; s)). \tag{9}$$

As $\alpha(t) \downarrow 0$, the optimization naturally shifts its focus from the Real+$D_\phi$ mixture to the $D_\phi$-only regime. According to the evaluation protocol, every $K$ steps, sample $P$ pairs of mirrored perturbations $\phi_j^{(\pm)} = \phi \pm \epsilon_j$. For each candidate, *run a brief parallel training trial* of $L$ update steps under the current $\alpha(t)$, then evaluate its return on $\mathcal{S}$ to obtain fitness values $F_j^{(\pm)}$. To reduce variance, each mirrored pair $\phi_j^{(+)}, \phi_j^{(-)}$ shares the same trial length $L$ and evaluation seed set $\mathcal{S}$ (while $\mathcal{S}$ can be periodically refreshed to mitigate overfitting to validation seeds).

**Mirrored ES Update & Guarantee.** Given these fitness values, we estimate the mirrored-ES gradient and update:

$$\hat{g} \;=\; \frac{1}{2P\sigma} \sum_{j=1}^{P} \left( F_j^{(+)} - F_j^{(-)} \right) \epsilon_j, \qquad \phi \leftarrow \mathrm{Proj}\big(\phi + \eta_{\mathrm{ES}}\,\hat{g}\big). \tag{10}$$

Here $\mathrm{Proj}(\cdot)$ enforces validity. To ensure data validity, we project the updated fields back to their feasible domains (e.g., clip continuous actions and round discrete values). Further details (including the parameterization of $\phi$ and the projection operators) are provided in the Appendix A.4.

**Theorem 1** (Approximate performance guarantee of SLED distillation). *Let $\pi_{SLED}$ be the final policy obtained in SLED when $\alpha(t) \to 0$ and ES converges to $\phi^\star$, trained on $D_{\phi^\star}$. Let $\pi_{real}$ be the optimal policy attainable with unlimited real data. Under standard regularity and Lipschitz assumptions detailed in the Appendix A.3, the performance gap admits the bound*

$$\big| J(\pi_{real}) - J(\pi_{SLED}) \big| \;\leq\; C_\gamma\big(\varepsilon_{\mathrm{approx}} + \varepsilon_{\mathrm{div}}(\phi^\star) + \varepsilon_{\mathrm{ES}}\big). \tag{11}$$

*Here $C_\gamma = \frac{c}{1-\gamma}$, where $c$ depends on the Lipschitz constant of the policy improvement operator and on the reward bound.*

The bound indicates that online-validated ES implicitly drives down the distributional mismatch term $\varepsilon_{\mathrm{div}}$, thereby supporting the final policy's performance.

---

**Algorithm 1** SLED (Self-Supervised Dataset Distillation)

---

1: **Init:** policy $\pi_\theta$, critic $Q_\psi$, buffer $\mathcal{B}$ (cap $C$), synthetic set $D_\phi$
2: **Schedules:** write $\beta(t)$, sample $\alpha(t)$; ES interval $K$, trial length $L$, pop $P$
3: **for** env step $t = 1$ to $T$ **do**
4:     Interact with env to get $x_t = (s, a, r, s', d)$
5:     **Write:** insert real with prob $\beta(t)$, else from $D_\phi$        (Eq. 2, 3)
6:     **Sample:** form batch quotas by $\alpha(t)$                (Eq. 5, 6)
7:     Training distribution $\pi_t(x) = \alpha(t)p_{\mathrm{real}} + (1 - \alpha(t))p_D$     (Eq. 7)
8:     **Update:** run base off-policy update on mixed batch (no PER/IS)
9:     **if** $t \bmod K = 0$ **then**
10:        **for** $j = 1..P$ **do**
11:           Sample mirrored $\phi_j^{(\pm)} = \phi \pm \epsilon_j$; run brief trial ($L$ steps), eval returns $F_j^{(\pm)}$
12:        **end for**
13:        $\hat{g} = \frac{1}{2P\sigma} \sum_{j=1}^{P}(F_j^{(+)} - F_j^{(-)})\epsilon_j$; $\phi \leftarrow \mathrm{Proj}\big(\phi + \eta_{\mathrm{ES}}\hat{g}\big)$    (Eq. 10)
14:     **end if**
15: **end for**

---

### 3.3 PLUG-AND-PLAY INTEGRATION AND OVERHEAD

**Integration.** Built on the two pillars introduced earlier, namely decoupled scheduling (§3.1) and online-validated ES (§3.2), SLED integrates as a cohesive drop-in module for standard off-policy algorithms such as DQN and TD3 without modifying their losses, targets, or update rules and without PER/IS. In practice, stratified mini-batches follow Eq. 5, with deficits in real samples backfilled according to Eq. 6 using $D_\phi$, while the critic and actor are updated using the original algorithm on the mixed batch. Every $K$ steps, the ES procedure in §3.2 runs asynchronously to refine $D_\phi$ through brief parallel training trials and a fixed evaluation protocol. In short, $\beta(t)$ governs what enters the buffer, $\alpha(t)$ governs what the learner consumes, and ES safeguards the quality of $D_\phi$.

**Overhead: theoretical comparison.** Let a transition occupy $b$ bytes. A standard ER buffer of capacity $C$ uses $M_{\text{ER}} = b\,C$. Under SLED, we maintain a unified FIFO buffer $B_t$ (cap. $C$) and a compact synthetic set $D_\phi$ of size $N \ll C$; the write-side probabilities satisfy $\Pr[x \in \text{real}] = \beta(t)$ and $\Pr[x \in D_\phi] = 1 - \beta(t)$ (Eq. (2)), and the FIFO recursion (Eq. (3)) implies that, as $\beta(t) \downarrow 0$, the buffer converges to $D_\phi$-only (Prop. 1). *Intuitively, the decoupled schedules let the learner consume a curriculum shaped by $\alpha(t)$ even while storage composition evolves via $\beta(t)$, front-loading a small, controllable distillation cost.* Hence the *effective training memory* (what the learner consumes as $\alpha(t) \downarrow 0$) satisfies $\lim_{t \to \infty} M_{\text{eff}}(t) = b\,N$, giving a reduction ratio $\Delta_{\text{mem}}^{\text{eff}} = 1 - \frac{N}{C}$. For compute, take one baseline mini-batch update as unit cost; SLED keeps the per-step learner update unchanged and adds only the ES routine with amortized cost $R_{\text{ES}} = \frac{PL}{K}$ (baseline-update equivalents per env step). Compared to common variants we have: ER (uniform) $\approx 1$; ER (PER) $\approx 1 + \kappa\,m \log C$ (batch size $m$, tree constant $\kappa$); HER (on-the-fly relabel $k$ goals) $\approx 1 + k$; SLED $\approx 1 + R_{\text{ES}}$. Therefore SLED is cheaper than PER when $R_{\text{ES}} < \kappa\,m \log C$ and cheaper than HER when $R_{\text{ES}} < k$; in the late phase $\alpha(t) \to 0$ and $\beta(t) \to 0$, sampling shifts from random access on $C$ to tensor slicing on $N \ll C$, further reducing access cost while ES is amortized by $(PL)/K$. In summary, SLED trades a bounded $R_{\text{ES}}$ term for a large effective-memory reduction and faster late-phase sampling; with ES on a separate worker, the learner's per-step time matches the baseline while retaining the $\Delta_{\text{mem}}^{\text{eff}} = 1 - N/C$ advantage.

## 4 EXPERIMENTS

We evaluate the proposed Self-Supervised Dataset Distillation (SLED) framework across three representative benchmarks: DMControl Tassa et al. (2018), Atari, and Habitat. These environments were selected to systematically examine SLED's generality, sample efficiency, and integration capability across continuous control, discrete-action exploration, and vision-based navigation. All models are trained under identical conditions using a shared infrastructure. Full experimental configurations, hyperparameters, and implementation details are provided in Appendix B.

**Baselines.** We compare SLED against a suite of representative replay-based baselines that span both classical and recent advances in experience management. Deep Q-Network (DQN) Mnih et al. (2015) is a foundational value-based algorithm that introduced experience replay, sampling uniformly from a buffer to stabilize Q-learning with a frozen target network. Soft Actor-Critic (SAC) Haarnoja et al. (2018) serves as a standard off-policy baseline, using a large FIFO buffer for uniform transition sampling in continuous control. Prioritized Experience Replay (PER) Schaul et al. (2015) enhances sample efficiency by prioritizing transitions with large TD errors, assuming they yield more informative updates. Hindsight Experience Replay (HER) Andrychowicz et al. (2017) improves learning in sparse-reward environments by relabeling failed goals to create meaningful learning signals. ReLo Sujit et al. (2023) proposes a learnability-based ranking scheme that prioritizes transitions based on their ability to consistently reduce training loss. LaBER Lahire et al. (2022) introduces a large-batch replay strategy that reweights samples according to gradient norm, amplifying the signal from informative transitions. SynthER Lu et al. (2023) synthesizes new transitions using diffusion models trained on past experiences, enabling flexible augmentation without extra environment interaction.

### 4.1 CONTINUOUS CONTROL IN DMCONTROL

We conduct an extensive evaluation on eight challenging tasks from the DeepMind Control Suite (DMControl) Tassa et al. (2018), where all methods are trained for 4 million environment steps. The results, presented in Fig. 2, demonstrate that SLED (red line) consistently achieves superiority in both sample efficiency and final performance. Specifically, in high-difficulty locomotion tasks such as `cheetah-run`, `walker-run`, and `finger-spin`, SLED not only learns faster but also surpasses all baselines in final performance. In the majority of the remaining tasks, including `fish-swim` and `acrobot-swingup`, SLED also performs within the top tier, significantly outperforming classic methods like PER and HER .These findings strongly validate SLED as a powerful and general-purpose framework. Through efficient generation and compression of synthetic experience, it achieves state-of-the-art performance and efficiency across a wide range of continuous control problems.

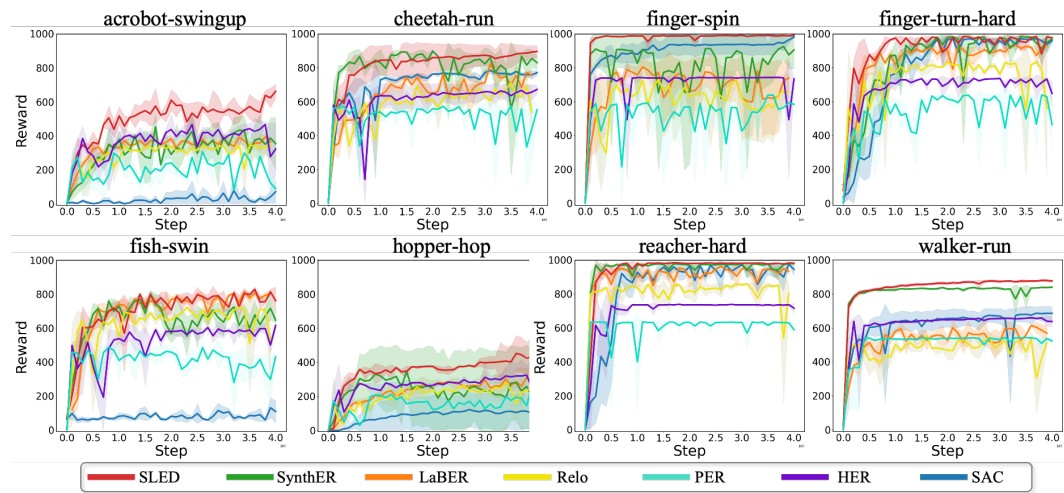

Figure 2: Average episode return on DMControl over 4M steps

## 4.2 EXPLORATION IN ATARI GAMES

We evaluate SLED on 7 Atari games Mnih et al. (2013), using DQN Mnih et al. (2015) as the base algorithm with a 1M replay buffer and 2M training steps. SLED maintains a 10k-size synthetic dataset, updated every 20k steps via Evolutionary Strategies. As shown in Table 1, SLED consistently outperforms all baselines, especially in exploration-heavy games like Frostbite. Despite using only 1% of the replay size, the distilled set captures high-utility transitions that improve credit assignment and generalization. These results demonstrate SLED's effectiveness in retaining meaningful experiences under sparse and noisy feedback.

Table 1: Scores comparison on 7 Atari games (averaged over 5 seeds).

| Method | Alien | Asterix | Breakout | Freeway | Qbert | MsPacman | Frostbite |
|--------|-------|---------|----------|---------|-------|----------|-----------|
| DQN | 1,721.2 | 4,274.6 | 10.1 | 30.6 | 13,127.3 | 5,840.3 | 3,025.6 |
| + PER | 4,204.2 | 31,527.3 | 14.0 | 33.7 | 16,256.5 | 6,519.1 | 4,380.1 |
| + LaBER | 4,365.2 | 39,172.1 | 15.4 | 31.6 | 17,744.6 | 6,691.4 | 4,923.5 |
| + ReLo | 4,312.9 | 38,432.4 | 16.0 | 37.6 | 19,013.2 | 6,613.1 | 4,892.7 |
| + SynthER | 4,203.5 | 39,521.4 | 17.0 | 37.6 | 19,192.2 | 6,683.1 | 4,992.7 |
| **+ SLED** | **4,625.4** | **39,612.7** | **18.7** | **41.3** | **19,432.3** | **6,831.2** | **5,232.5** |

## 4.3 VISUAL NAVIGATION IN HABITAT

We further evaluate SLED on high-dimensional visual navigation in AI Habitat Savva et al. (2019); Szot et al. (2021); Puig et al. (2023). Using SAC as the base learner, we compare against PER, HER, and SynthER across three HM3D scenes Ramakrishnan et al. (2021), Residential, Office, and Commercial, with agents observing egocentric RGB-D only; success is defined as reaching the goal within a fixed step budget. Across 100 evaluation episodes per scene, SLED achieves the highest success rates in all settings, exceeding the next best by more than 3.5% in the most challenging Commercial scene (Table 2).

Table 2: Success rates (%) on Habitat navigation tasks.

| Method | Residential | Office | Commercial |
|--------|-------------|--------|------------|
| SAC | 31.7 ± 5.1 | 38.1 ± 1.9 | 42.2 ± 2.5 |
| + PER | 48.0 ± 3.1 | 52.3 ± 2.8 | 51.7 ± 1.5 |
| + HER | 50.2 ± 2.9 | 56.4 ± 2.6 | 53.0 ± 2.7 |
| + SynthER | 54.3 ± 2.6 | 59.1 ± 2.1 | 57.8 ± 2.0 |
| **+ SLED** | **63.1 ± 2.1** | **66.1 ± 1.4** | **63.5 ± 1.9** |

### 4.4 ABLATION STUDIES

We evaluate the contributions of SLED's components on `ALE/Breakout-v5` over 2M steps, building upon a standard DQN agent. We compare the full model with three variants: **w/o decoupling**, which couples the sampling and writing schedules ($\alpha(t) = \beta(t)$); **w/o ES optimization**, which fixes the synthetic dataset $D_\phi$ after initialization; and **w/o synthetic dataset**,which effectively reverts to the baseline DQN. As shown in Table 3, we measure performance using the **Final Score** and early sample efficiency (**Score/200k**), alongside costs like **Total Time** and peak GPU memory (**Peak (MiB)**). The results show that while removing the decoupled schedules causes severe performance degradation, the removal of ES optimization is even more detrimental, underscoring the critical roles of both components. Furthermore, while the synthetic dataset and its ES optimization introduce a minimal time overhead (+4 minutes), they are essential for achieving a higher final return and superior sample efficiency.

Table 3: Ablation study of SLED components on `ALE/Breakout-v5` over 2M steps.

| Ablation Variant | Final Score | Score/200k | Time | Peak (MiB) | N/C (%) | $R_{\mathrm{ES}}$ |
|---|---|---|---|---|---|---|
| SLED (Full) | **18.7** | **9.1** | 01:34 | 189 | 1% | 0.04 |
| w/o decoupling | 10.5 | 5.2 | 01:31 | 189 | 1% | 0.04 |
| w/o ES optimization | 8.8 | 4.9 | 01:32 | **183** | 1% | - |
| w/o synthetic dataset | 12.1 | 5.3 | **01:30** | 466 | - | - |

## 5 RELATED WORK

**Experience Replay.** Experience Replay (ER) is a foundational mechanism in off-policy reinforcement learning that improves sample efficiency by storing and reusing past transitions Lin (1992); Mnih et al. (2013); Yang et al. (2024). Prioritized Experience Replay (PER)Schaul et al. (2015) improves upon uniform sampling by favoring transitions with high TD error, leading to faster convergence. Subsequent works expand PER by incorporating additional sampling heuristics, such as topological structureHong et al. (2022), replay frequency Wei et al. (2021), and feature-level similarity Yu et al. (2024); Yang et al. (2023b). Hindsight Experience Replay (HER)Andrychowicz et al. (2017) enables goal-conditioned agents to learn from failure by relabeling goals post hoc, and has been extended through curriculum learningFang et al. (2019) and decomposition-based relabeling Luo et al. (2023); Zhao et al. (2025). Distributed variants like Ape-X Horgan et al. (2018); Yang et al. (2023a) and IMPALA Espeholt et al. (2018) further improve scalability. ER is effective but relies on massive buffers, raising costs and reducing sample relevance.

**Dataset Distillation.** Dataset distillation aims to learn a compact synthetic dataset that can approximate the training utility of a much larger real dataset Wang et al. (2018). Most early works apply gradient-matching techniques in supervised learning settings Zhao et al. (2021), while recent extensions explore broader objectives such as feature alignment Cazenavette & Eriksson (2022) or trajectory consistency Lee et al. (2022). In reinforcement learning, dataset distillation has been adapted for experience replay compression and continual learning. DREAM Liu et al. (2022) distills Q-learning transitions by matching the value gradients over synthetic and real samples. SLDR Zheng et al. (2023) co-optimizes a small set of support trajectories alongside the policy. DIET Hu et al. (2023) introduces distillation for multi-task RL with memory constraints. SLED applies gradient-free ES distillation, avoiding differentiable-target limits and base modifications in non-stationary RL.

## 6 CONCLUSION

We introduced SLED, a plug-and-play framework that replaces large replay buffers with a compact synthetic dataset by *decoupling* buffer composition $\beta(t)$ from the training curriculum $\alpha(t)$. An online-validated ES routine, using brief parallel training trials and mirrored perturbations, directly optimizes downstream return, while leaving base actor–critic updates unchanged and avoiding PER/IS. Theoretically, the performance gap is bounded by a term scaling with $C_\gamma = \frac{c}{1-\gamma}$; empirically, SLED improves sample efficiency and convergence with a tiny memory footprint as the buffer converges to distilled-only storage. Overall, SLED offers a lightweight, scalable path to memory-efficient off-policy RL.

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

## A  THEORETICAL ANALYSIS AND PROOFS

**Notation.**  Let $\gamma \in (0,1)$ and $C_\gamma \triangleq \frac{1}{1-\gamma}$. For any policy $\pi$, let $d_\pi$ be its $\gamma$-discounted state-visitation distribution and $Q_\pi$ its action-value. Rewards are bounded by $|r| \leq R_{\max}$ so that $\|Q_\pi\|_\infty \leq R_{\max} C_\gamma$. Throughout, $\|\cdot\|_1$ and $\|\cdot\|_\infty$ denote $L_1$ and sup norms, respectively. We reuse equation labels from the main text when referenced (e.g., Eq. 2, Eq. 3, Eq. 5, Eq. 6, Eq. 7).

### A.1  PROOF OF PROPOSITION 1 (ASYMPTOTIC BUFFER COMPOSITION)

**Setup and notation.**  Let $X_t \in \{0, 1, \ldots, C\}$ denote the (random) number of real items in the buffer at step $t$ and $\rho_{\text{real}}(t) \triangleq \mathbb{E}[X_t]/C$ the *expected* fraction of real items. At step $t$, the incoming item is real with probability $\beta(t)$ and synthetic (from $D_\phi$) with probability $1 - \beta(t)$, cf. Eq. 2. We analyze the steady-state (full-buffer) regime; before the buffer is full ($t < C$) one obtains the same conclusion with a slightly different warm-up recursion (see Remark A.1).

**Step 1 – One-step recursion in expectation.** When the buffer is full, one oldest item is evicted and one new item is written each step. Conditioning on $X_t$ and using the law of total expectation yields

$$\mathbb{E}[X_{t+1} \mid X_t] = X_t - \tfrac{1}{C} X_t + \beta(t), \qquad \Rightarrow \qquad \rho_{\text{real}}(t+1) = \left(1 - \tfrac{1}{C}\right)\rho_{\text{real}}(t) + \tfrac{1}{C}\beta(t), \quad (12)$$

which coincides with Eq. 3. This is an affine, time-varying, first-order stable filter.

**Step 2 – Closed form (exponential moving average).** Unrolling Eq. 12 gives, for any $\rho_{\text{real}}(0) \in [0, 1]$,

$$\rho_{\text{real}}(t) = \left(1 - \tfrac{1}{C}\right)^t \rho_{\text{real}}(0) + \tfrac{1}{C} \sum_{k=0}^{t-1} \left(1 - \tfrac{1}{C}\right)^k \beta(t - 1 - k). \tag{13}$$

Thus $\rho_{\text{real}}$ is an exponentially weighted moving average (EWMA) of $\beta(\cdot)$.

**Step 3 – Limit as $\beta(t) \to 0$.** If $\lim_{t\to\infty} \beta(t) = 0$ and $0 < 1 - \tfrac{1}{C} < 1$, then (i) the homogeneous term in Eq. 13 vanishes; (ii) the inhomogeneous term is a convolution of a summable geometric kernel with a sequence converging to $0$ and therefore vanishes by Toeplitz's lemma. Hence

$$\lim_{t\to\infty} \rho_{\text{real}}(t) = 0,$$

i.e., the buffer converges *in expectation* to $D_\phi$-only storage.

**Step 4 – Parallel writers (generalization).** If an average of $\kappa \in (0, C]$ items are written per step (e.g., $E$ parallel envs), the recursion becomes

$$\rho_{\text{real}}(t+1) = \left(1 - \tfrac{\kappa}{C}\right)\rho_{\text{real}}(t) + \tfrac{\kappa}{C}\beta(t),$$

and all arguments go through with $1/C$ replaced by $\kappa/C$.

**Variance and concentration (optional).**  Let the *random* real fraction be $\widehat{\rho}_{\text{real}}(t) \triangleq X_t/C$ (so that $\mathbb{E}[\widehat{\rho}_{\text{real}}(t)] = \rho_{\text{real}}(t)$). Write

$$X_{t+1} = X_t - Y_t + B_t,$$

where $Y_t \mid X_t \sim \text{Bernoulli}(X_t/C)$ (eviction) and $B_t \sim \text{Bernoulli}(\beta(t))$ (write), with $B_t \perp (X_t, Y_t)$. Then

$$\begin{aligned} \text{Var}(X_{t+1}) &= \text{Var}(X_t - Y_t) + \text{Var}(B_t) \\ &= \text{Var}(X_t) + \text{Var}(Y_t) - 2\,\text{Cov}(X_t, Y_t) + \text{Var}(B_t). \end{aligned} \tag{14}$$

Since $\mathbb{E}[Y_t \mid X_t] = X_t/C$,

$$\text{Cov}(X_t, Y_t) = \text{Cov}\big(X_t, \mathbb{E}[Y_t \mid X_t]\big) = \tfrac{1}{C}\text{Var}(X_t). \tag{15}$$

Moreover, with $p_t \triangleq X_t/C$,

$$\begin{aligned} \text{Var}(Y_t) &= \mathbb{E}\big[\text{Var}(Y_t \mid X_t)\big] + \text{Var}\big(\mathbb{E}[Y_t \mid X_t]\big) \\ &= \mathbb{E}[p_t(1 - p_t)] + \text{Var}(p_t) = \mathbb{E}[p_t] - \mathbb{E}[p_t]^2 = \frac{\mathbb{E}[X_t]}{C} - \frac{(\mathbb{E}[X_t])^2}{C^2} \leq \tfrac{1}{4}, \end{aligned} \tag{16}$$

where the last inequality is maximized at $\mathbb{E}[X_t]/C = 1/2$. Combining Eq. 14–Eq. 16 and using $\mathrm{Var}(B_t) \leq \frac{1}{4}$,

$$\mathrm{Var}(X_{t+1}) \ \leq \ \left(1 - \tfrac{2}{C}\right)\mathrm{Var}(X_t) + \tfrac{1}{4} + \mathrm{Var}(B_t) \ \leq \ \left(1 - \tfrac{2}{C}\right)\mathrm{Var}(X_t) + \tfrac{1}{2}. \tag{17}$$

Dividing Eq. 17 by $C^2$ gives

$$\mathrm{Var}\big(\widehat{\rho}_{\mathrm{real}}(t{+}1)\big) \ \leq \ \left(1 - \tfrac{2}{C}\right)\mathrm{Var}\big(\widehat{\rho}_{\mathrm{real}}(t)\big) + \frac{1}{2C^2}. \tag{18}$$

Hence, in steady state,

$$\mathrm{Var}\big(\widehat{\rho}_{\mathrm{real}}\big) \ \leq \ \frac{1}{4C}, \qquad \text{and fluctuations are} \ \ \mathrm{sd}\big(\widehat{\rho}_{\mathrm{real}}\big) = O\big(C^{-1/2}\big),$$

matching the empirical stability for large $C$. Since $|X_{t+1} - X_t| \leq 1$ almost surely in the full-buffer regime, a standard Azuma–Hoeffding argument yields the same $O(C^{-1/2})$ fluctuation scale for $\widehat{\rho}_{\mathrm{real}}(t)$.

**Remark (warm-up).** Before the buffer is full,

$$\rho_{\mathrm{real}}(t{+}1) = \tfrac{t}{t+1}\,\rho_{\mathrm{real}}(t) + \tfrac{1}{t+1}\,\beta(t),$$

the usual online average. After $t \geq C$, the FIFO recursion Eq. 12 applies; both regimes yield the same limit when $\beta(t) \to 0$.

---

## A.2    PROOF OF PROPOSITION 2 (TV BOUND FOR CURRICULUM DEVIATION)

Let the *target* training distribution at step $t$ be

$$\pi_t(x) = \alpha(t)\,p_{\mathrm{real}}(x) + \big(1 - \alpha(t)\big)\,p_D(x),$$

as in Eq. 7. Quota sampling with batch size $m$ draws $m_{\mathrm{real}}(t) = \lfloor m\,\alpha(t) \rfloor$ from the buffer and $m_D(t) = m - m_{\mathrm{real}}(t)$ from $D_\phi$ (without replacement), with backfilling per Eq. 6 if real items are insufficient. Let $\hat{p}_{\mathrm{real}}$ and $\hat{p}_D$ be the empirical sub-distributions, and define

$$\hat{\pi}_t \ \triangleq \ \hat{\alpha}_t\,\hat{p}_{\mathrm{real}} + \big(1 - \hat{\alpha}_t\big)\,\hat{p}_D, \qquad \hat{\alpha}_t \ \triangleq \ \frac{m_{\mathrm{real}}^\star(t)}{m} \in [0,1].$$

**Step 1 – TV convexity and positive homogeneity.** Total variation satisfies $\mathrm{TV}(p,q) = \frac{1}{2}\|p - q\|_1$ and is convex and positively homogeneous. With the same-weight auxiliary mixture $\tilde{\pi}_t \triangleq \alpha(t)\,\hat{p}_{\mathrm{real}} + (1 - \alpha(t))\,\hat{p}_D$, we have

$$\mathrm{TV}(\hat{\pi}_t, \pi_t) \leq \mathrm{TV}(\hat{\pi}_t, \tilde{\pi}_t) + \mathrm{TV}(\tilde{\pi}_t, \pi_t) \tag{19}$$

$$= \tfrac{1}{2}\big\|(\hat{\alpha}_t - \alpha(t))(\hat{p}_{\mathrm{real}} - \hat{p}_D)\big\|_1 + \tfrac{1}{2}\big\|\alpha(t)(\hat{p}_{\mathrm{real}} - p_{\mathrm{real}}) + (1 - \alpha(t))(\hat{p}_D - p_D)\big\|_1$$

$$\leq \big|\hat{\alpha}_t - \alpha(t)\big| + \alpha(t)\,\mathrm{TV}(\hat{p}_{\mathrm{real}}, p_{\mathrm{real}}) + \big(1 - \alpha(t)\big)\,\mathrm{TV}(\hat{p}_D, p_D). \tag{20}$$

This yields the refined bound

$$\boxed{\mathrm{TV}(\hat{\pi}_t, \pi_t) \ \leq \ \alpha(t)\,\mathrm{TV}(\hat{p}_{\mathrm{real}}, p_{\mathrm{real}}) + \big(1 - \alpha(t)\big)\,\mathrm{TV}(\hat{p}_D, p_D) + \big|\hat{\alpha}_t - \alpha(t)\big|} \tag{21}$$

**Step 2 – Rounding/backfilling correction.** If the real stratum is sufficient, then

$$\big|\hat{\alpha}_t - \alpha(t)\big| = \left|\frac{\lfloor m\alpha(t) \rfloor}{m} - \alpha(t)\right| \leq \tfrac{1}{m}.$$

Under backfilling (real scarce), $\hat{\alpha}_t = N_{\mathrm{real}}(t)/m \leq m_{\mathrm{real}}(t)/m$ and

$$\big|\hat{\alpha}_t - \alpha(t)\big| \leq \tfrac{m_{\mathrm{real}}(t) - N_{\mathrm{real}}(t)}{m},$$

which vanishes once real items are replenished (and trivially as $\alpha(t) \downarrow 0$). If one prefers the minimal statement of Proposition 2, simply drop the nonnegative weight term in Eq. 21 to obtain

$$\mathrm{TV}(\hat{\pi}_t, \pi_t) \leq \alpha(t)\,\mathrm{TV}(\hat{p}_{\mathrm{real}}, p_{\mathrm{real}}) + \big(1 - \alpha(t)\big)\,\mathrm{TV}(\hat{p}_D, p_D). \tag{22}$$

**Step 3 – Finite-sample effect (remark).** Within each stratum, without-replacement sampling enjoys the same first-order concentration as i.i.d. sampling; hence

$$\mathbb{E}\,\mathrm{TV}(\hat{p}_{\mathrm{real}}, p_{\mathrm{real}}) = O\big(m_{\mathrm{real}}^{-1/2}\big), \qquad \mathbb{E}\,\mathrm{TV}(\hat{p}_D, p_D) = O\big(m_D^{-1/2}\big),$$

so $\mathbb{E}\,\mathrm{TV}(\hat{\pi}_t, \pi_t) = O(m^{-1/2})$ and the rounding term contributes at most $O(1/m)$ when the real stratum is sufficient. $\qquad\square$

---

## A.3 PROOF OF THEOREM 1 (APPROXIMATE PERFORMANCE GUARANTEE)

**Standing assumptions.** (i) Rewards are bounded: $|r| \leq R_{\max}$, hence $\|Q_\pi\|_\infty \leq R_{\max}/(1-\gamma)$. (ii) The base policy-improvement operator is Lipschitz (the constant is absorbed into $c$ below). (iii) ES converges to a stationary point $\phi^\star$ up to a finite-sample error summarized by $\varepsilon_{\mathrm{ES}}$ (see Step 4).

Let $\pi_{\mathrm{real}}$ denote the policy attainable with unlimited real data, and let $\pi_{\mathrm{SLED}}$ be the policy trained on $D_{\phi^\star}$ when $\alpha(t) \to 0$.

**Goal.** Show the *linear-in-*$1/(1-\gamma)$ return-gap bound

$$\big|J(\pi_{\mathrm{real}}) - J(\pi_{\mathrm{SLED}})\big| \leq \frac{c}{1-\gamma}\Big(\varepsilon_{\mathrm{approx}} + \varepsilon_{\mathrm{div}}(\phi^\star) + \varepsilon_{\mathrm{ES}}\Big), \tag{23}$$

where $\varepsilon_{\mathrm{approx}}$ covers function-approximation / finite-training error of the base learner, and the value-weighted divergence term is

$$\varepsilon_{\mathrm{div}}(\phi^\star) \triangleq \mathbb{E}_{s \sim d_{\pi_{\mathrm{SLED}}}}\left[\sum_a \big|\pi_{\mathrm{real}}(a|s) - \pi_{\mathrm{SLED}}(a|s)\big| \cdot |Q_{\pi_{\mathrm{real}}}(s,a)|\right]. \tag{24}$$

**Step 1 – Symmetric performance-difference lemma.** We use the symmetric form (see, e.g., TRPO / Kakade–Langford):

$$J(\pi) - J(\pi') = \frac{1}{1-\gamma}\,\mathbb{E}_{s \sim d_{\pi'}}\left[\sum_a \big(\pi(a|s) - \pi'(a|s)\big)\, Q_\pi(s,a)\right]. \tag{25}$$

Setting $\pi = \pi_{\mathrm{real}}$ and $\pi' = \pi_{\mathrm{SLED}}$ gives

$$J(\pi_{\mathrm{real}}) - J(\pi_{\mathrm{SLED}}) = \frac{1}{1-\gamma}\,\mathbb{E}_{s \sim d_{\pi_{\mathrm{SLED}}}}\left[\sum_a \big(\pi_{\mathrm{real}} - \pi_{\mathrm{SLED}}\big)\, Q_{\pi_{\mathrm{real}}}\right].$$

**Step 2 – Value-weighted policy gap.** By the triangle inequality,

$$\big|J(\pi_{\mathrm{real}}) - J(\pi_{\mathrm{SLED}})\big| \leq \frac{1}{1-\gamma}\,\mathbb{E}_{s \sim d_{\pi_{\mathrm{SLED}}}}\left[\sum_a \big|\pi_{\mathrm{real}}(a|s) - \pi_{\mathrm{SLED}}(a|s)\big|\,|Q_{\pi_{\mathrm{real}}}(s,a)|\right]$$

$$= \frac{1}{1-\gamma}\,\varepsilon_{\mathrm{div}}(\phi^\star). \tag{26}$$

This is the core value-weighted curriculum-mismatch term evaluated under the *final* state distribution $d_{\pi_{\mathrm{SLED}}}$, hence no extra change-of-measure penalty appears.

**Step 3 – Base-learner approximation error.** Let $\varepsilon_{\mathrm{approx}}$ bound the residual due to function approximation, finite training, and algorithmic stochasticity: the policy produced by the base learner from a given data distribution is $L$-Lipschitz (in the appropriate metric) with respect to its inputs,

and the resulting performance error contribution scales as $\frac{c_{\mathrm{approx}}}{1-\gamma}\,\varepsilon_{\mathrm{approx}}$ for some constant $c_{\mathrm{approx}}$ absorbed into $c$.

**Step 4 – ES suboptimality.** The mirrored-ES estimator is unbiased and has variance $\tilde{O}(d_\phi/(P\sigma^2))$; finite population $P$, trial length $L$, and projection onto feasible domains yield an optimization sub-optimality $\varepsilon_{\mathrm{ES}}$, whose contribution to the return scales at most linearly with $1/(1-\gamma)$: $\frac{c_{\mathrm{ES}}}{1-\gamma}\,\varepsilon_{\mathrm{ES}}$, with $c_{\mathrm{ES}}$ absorbed into $c$.

**Step 5 – Combine (linear scaling in $1/(1-\gamma)$).** Adding Eq. 26–Step 4 and gathering constants gives

$$\boxed{\left| J(\pi_{\mathrm{real}}) - J(\pi_{\mathrm{SLED}}) \right| \;\leq\; \frac{c}{1-\gamma}\Big( \varepsilon_{\mathrm{div}}(\phi^\star) + \varepsilon_{\mathrm{approx}} + \varepsilon_{\mathrm{ES}} \Big)} \tag{27}$$

where $c$ depends only on reward bounds and Lipschitz constants (and is independent of $\gamma$). Hence the return-gap exhibits a *linear* dependence on $1/(1-\gamma)$, as claimed. $\qquad\square$

**Remark (on constants).** If desired, one can make $c$ explicit; for instance, with $|r| \leq R_{\max}$ and a base-learner Lipschitz constant $L_{\mathrm{pl}}$, one may take $c = 1 + L_{\mathrm{pl}} + c_{\mathrm{ES}}$, which is independent of $\gamma$. The factor $1/(1-\gamma)$ in Eq. 27 is therefore the *only* explicit dependence on the horizon, yielding the stated linear scaling.

---

## A.4 Properties of the Mirrored-ES Gradient Estimator

We formalize the estimator used in the ES update Eq. 10. Let $F(\phi)$ denote the black-box objective produced by the trial-and-evaluate pipeline in Eq. 9, and define its Gaussian-smoothed version

$$\mathcal{F}_\sigma(\phi) \;\triangleq\; \mathbb{E}_{\epsilon\sim\mathcal{N}(0,\sigma^2 I)}[F(\phi+\epsilon)]\,.$$

Unless stated otherwise, expectations are w.r.t. the perturbations $\epsilon$. Assume the objective is uniformly bounded on the feasible domain: $|F(\phi)| \leq W_{\max}$.

At ES step $t$, draw i.i.d. $\epsilon_1,\ldots,\epsilon_P \sim \mathcal{N}(0,\sigma^2 I_{d_\phi})$ and form mirrored candidates $\phi_j^{(\pm)} = \phi \pm \epsilon_j$. Let $F_j^{(\pm)} \equiv F(\phi_j^{(\pm)})$, evaluated with *common random numbers* (CRN) within each pair (same trial length and the same evaluation seed set $\mathcal{S}$). The estimator used in the main text is

$$\hat{g}_{\mathrm{mir}} \;=\; \frac{1}{2P\sigma} \sum_{j=1}^{P} \big(F_j^{(+)} - F_j^{(-)}\big)\,\epsilon_j, \qquad \phi \leftarrow \mathrm{Proj}\big(\phi + \eta_{\mathrm{ES}}\hat{g}_{\mathrm{mir}}\big). \tag{10}$$

**Main facts.** With the setup above, $\hat{g}_{\mathrm{mir}}$ satisfies

$$\textbf{(i) Unbiasedness:} \qquad \mathbb{E}[\hat{g}_{\mathrm{mir}}] \;=\; \nabla_\phi \mathcal{F}_\sigma(\phi). \tag{28}$$

$$\textbf{(ii) Second moment and variance:} \qquad \mathbb{E}\big[\|\hat{g}_{\mathrm{mir}}\|_2^2\big] \;\leq\; \frac{c_m\, d_\phi}{P\,\sigma^2}\, W_{\max}^2 \;+\; \frac{d_\phi}{\sigma^2}\, W_{\max}^2, \tag{29}$$

$$\mathrm{Var}\big(\hat{g}_{\mathrm{mir}}\big) \;\leq\; \frac{C_m\, d_\phi}{P\,\sigma^2}\, W_{\max}^2 \tag{30}$$

for absolute constants $c_m, C_m$. When CRN is used within each mirrored pair, one can take $c_m = 1$ and $C_m = 2$ (up to small constant slack), hence $\mathrm{Var}(\hat{g}_{\mathrm{mir}}) = \tilde{O}\big(d_\phi/(P\sigma^2)\big)$.

**Proof of Eq. 28.** By Gaussian symmetry $\epsilon \overset{d}{=} -\epsilon$,

$$\mathbb{E}\big[(F(\phi+\epsilon) - F(\phi-\epsilon))\,\epsilon\big] = 2\,\mathbb{E}[F(\phi+\epsilon)\,\epsilon]\,.$$

By the multivariate Stein identity, for any smooth $f$ and $\epsilon \sim \mathcal{N}(0,\sigma^2 I)$, $\mathbb{E}[f(\epsilon)\,\epsilon] = \sigma^2\,\mathbb{E}[\nabla f(\epsilon)]$. Taking $f(\epsilon) = F(\phi+\epsilon)$ gives $\mathbb{E}[F(\phi+\epsilon)\epsilon] = \sigma^2 \nabla_\phi \mathbb{E}[F(\phi+\epsilon)] = \sigma^2 \nabla_\phi \mathcal{F}_\sigma(\phi)$. Divide by $2\sigma$ and average over $P$ i.i.d. pairs to obtain $\mathbb{E}[\hat{g}_{\mathrm{mir}}] = \nabla_\phi \mathcal{F}_\sigma(\phi)$.

**Proof of Eq. 30.** Let $Z_j = \frac{1}{2\sigma}\big(F(\phi + \epsilon_j) - F(\phi - \epsilon_j)\big)\epsilon_j$, so $\hat{g}_{\mathrm{mir}} = \frac{1}{P}\sum_{j=1}^{P} Z_j$ with $\mathbb{E}[Z_j] = \nabla_\phi \mathcal{F}_\sigma(\phi)$. Using $|F| \leq W_{\max}$ and $\mathbb{E}\|\epsilon\|_2^2 = d_\phi \sigma^2$,

$$\mathbb{E}\|Z_1\|_2^2 = \frac{1}{4\sigma^2}\,\mathbb{E}\big[(F(\phi + \epsilon) - F(\phi - \epsilon))^2\|\epsilon\|_2^2\big] \;\leq\; \frac{(2W_{\max})^2}{4\sigma^2}\,d_\phi \sigma^2 \;=\; d_\phi W_{\max}^2,$$

and with CRN the constant improves (captured by $c_m$). By independence,

$$\mathbb{E}\|\hat{g}_{\mathrm{mir}}\|_2^2 = \frac{1}{P}\,\mathbb{E}\|Z_1\|_2^2 + \frac{P-1}{P}\,\big\|\mathbb{E}Z_1\big\|_2^2 \;\leq\; \frac{c_m\, d_\phi}{P}\,W_{\max}^2 + \big\|\nabla_\phi \mathcal{F}_\sigma(\phi)\big\|_2^2.$$

Finally, $\|\nabla_\phi \mathcal{F}_\sigma(\phi)\|_2 \leq \frac{1}{\sigma}\,\mathbb{E}[|F(\phi + \epsilon)|\,\|\epsilon\|_2] \leq \frac{W_{\max}}{\sigma}\sqrt{d_\phi}$, so $\|\nabla_\phi \mathcal{F}_\sigma(\phi)\|_2^2 \leq \frac{d_\phi}{\sigma^2}W_{\max}^2$, which yields the stated second-moment bound; the variance bound follows from $\mathrm{Var}(\hat{g}_{\mathrm{mir}}) = \frac{1}{P}\mathrm{Var}(Z_1)$ and the estimate $\mathrm{Var}(Z_1) \leq C_m\,\frac{d_\phi}{\sigma^2}W_{\max}^2$.

**Remark (projection).** The update $\mathrm{Proj}(\phi + \eta_{\mathrm{ES}}\hat{g}_{\mathrm{mir}})$ enforces feasibility (e.g., clipping continuous actions, rounding discrete fields). This projection does not affect the unbiasedness of $\hat{g}_{\mathrm{mir}}$ for $\nabla \mathcal{F}_\sigma(\phi)$; it only constrains the optimization path.

**Remark (bias vs. the unsmoothed objective).** Both one-sided and mirrored ES estimators are unbiased for $\nabla \mathcal{F}_\sigma(\phi)$, the gradient of the Gaussian-smoothed objective. If $F$ is $L$-Lipschitz (or sufficiently smooth), the smoothing bias $\|\nabla \mathcal{F}_\sigma(\phi) - \nabla F(\phi)\|$ is $O(\sigma)$, so decreasing $\sigma$ trades bias for variance in the usual way.

## B  BENCHMARK ENVIRONMENTS

We evaluate SLED across three representative benchmarks, *DMControl*, *Atari*, and *Habitat*, chosen to probe generality across continuous control, discrete action exploration, and vision-based navigation. Together, these suites span low-dimensional proprioceptive control, high-dimensional pixel-based decision making, and photorealistic embodied navigation. Unless otherwise specified, results are averaged over 5 random seeds.

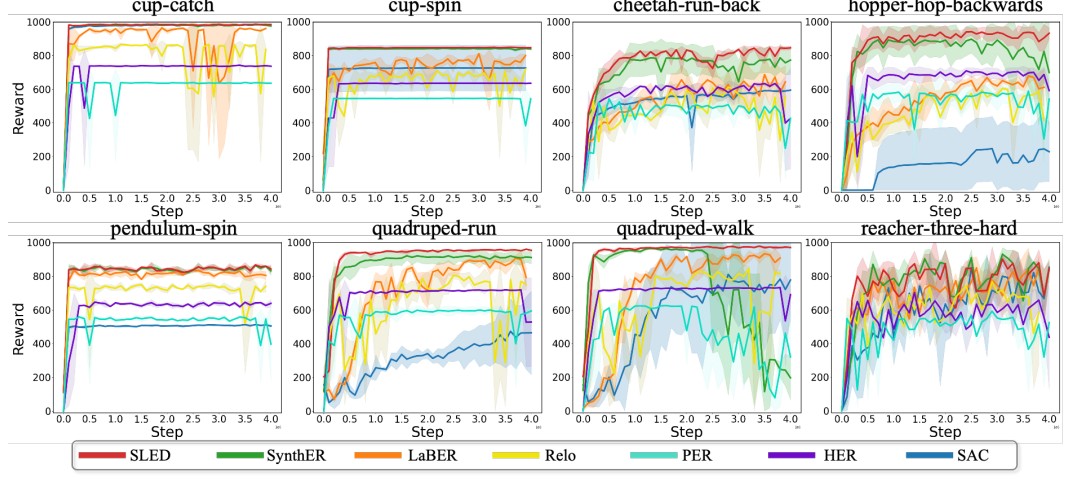

Figure 3: Average episode return on DMControl over 4M steps

**DeepMind Control Suite (DMControl).** The DMControl benchmark (Tassa et al., 2018) provides a standardized set of physics-based continuous control tasks with MuJoCo dynamics. Besides the main experiments, additional results are shown in Figure 3. We select eight challenging environments that cover locomotion, underactuated dynamics, and sparse manipulation:

- **Acrobot-Swingup** (obs: 6, act: 1) – a two-link underactuated pendulum where rewards only trigger when the tip is swung upright, testing long-horizon credit assignment.
- **Cheetah-Run** (obs: 17, act: 6) – periodic high-speed locomotion that requires coordinating multiple joints for forward velocity.
- **Finger-Spin** (obs: 9, act: 2, sparse) – rotate a free-floating spinner continuously with sparse binary rewards; exploration is particularly difficult.
- **Finger-Turn-Hard** (obs: 12, act: 2, sparse) – turn a spinner to match a precise goal angle under sparse feedback, stressing precision control.
- **Fish-Swim** (obs: 24, act: 5) – locomotion in fluid-like dynamics with coupled joints, demanding smooth coordination.
- **Hopper-Hop** (obs: 15, act: 4) – single-legged hopping, highly unstable and sensitive to balance errors.
- **Reacher-Hard** (obs: 6, act: 2, sparse) – reaching distant planar targets with very sparse reward signals.
- **Walker-Run** (obs: 24, act: 6) – bipedal locomotion where stability and gait coordination are required over long horizons.

Observations are low-dimensional proprioceptive state vectors; actions are continuous torques. Each episode is capped at 1,000 steps, with agents trained for **4M environment steps**. These tasks expose both dense and sparse reward structures, under which SLED shows consistent sample efficiency improvements (Fig. 2).

**Atari Arcade Learning Environment.**    The Atari 2600 suite (Mnih et al., 2013) is a long-standing benchmark for pixel-based reinforcement learning. We use DQN (Mnih et al., 2015) as the base learner, with a **1M** FIFO buffer and **2M** training steps. Observations are $84 \times 84$ grayscale frames with 4-frame stacking; actions are discrete joystick commands. We evaluate on seven diverse games:

- **Alien** – sparse rewards requiring exploration across large maps with many distractors.
- **Asterix** – fast-paced environment demanding precise timing and short-term credit assignment.
- **Breakout** – sparse rewards and high variance: learning depends on rare contact.
- **Freeway** – binary reward for lane crossing; exploration bottleneck due to feedback.
- **Q*bert** – complex rules and changing reward semantics across multiple levels.
- **MsPacman** – partially observable mazes with moving stochastic opponents, requiring long-term planning.
- **Frostbite** – highly exploration-heavy with sparse success conditions and stochastic hazards.

SLED maintains a compact synthetic dataset $D_\phi$ of 10k transitions, updated asynchronously every 20k steps via Evolutionary Strategies. Despite using only 1% of the replay size, SLED consistently outperforms PER, LaBER, ReLo, and SynthER (Table 1), especially on exploration-heavy games such as `Frostbite`.

**Habitat Visual Navigation.**    We further evaluate in AI Habitat (Savva et al., 2019; Szot et al., 2021; Puig et al., 2023), which provides photorealistic 3D indoor navigation tasks based on HM3D scenes (Ramakrishnan et al., 2021). Agents observe egocentric RGB-D frames ($256 \times 256$) and relative goal coordinates; the action space consists of discrete movements (forward, turn, strafe, stop). Success is defined as issuing a `STOP` action within **0.2 m** of the goal, under a cap of **1,000** steps. We evaluate three scene categories:

- **Residential** – cluttered homes with narrow corridors and frequent occlusions.
- **Office** – multi-room layouts with long hallways and repetitive visual textures.
- **Commercial** – wide, open layouts such as malls or stores, with diverse lighting and open spaces.

We train SAC as the base learner, comparing against SAC+PER, SAC+HER, and SynthER. Evaluation is averaged over **100 episodes per scene**. As reported in Table 2, SLED achieves the highest success rates across all settings, surpassing the next best method by more than 3.5% in the hardest `Commercial` scene, demonstrating its robustness under perceptual noise and complex spatial layouts.

Overall, these three benchmarks stress complementary challenges: exploration under sparse rewards (DMControl, Atari), sample efficiency with limited buffer size (Atari), and generalization under visual complexity (Habitat). Across all, SLED improves performance by refining the replay pipeline without altering the base learner's update rules.

## C   FURTHER ANALYSIS ON ATARI

We focus all analyses on `ALE/Breakout-v5` to isolate causal effects and to avoid cross-task confounds. Unless stated otherwise, we report the mean over 5 seeds. Beyond final score, we quantify (i) **early efficiency** (Score@200k), (ii) **learning stability** via catastrophic drops (CD; # of $\geq 40\%$ score collapses after 500k frames), and (iii) **area-under-curve** up to 1M frames (AUC@1M), which tracks how quickly usable performance accrues.

### C.1   ABLATIONS AS A FACTORIAL STUDY (DECOUPLING × ES)

Rather than ablating modules one-by-one, we run a $2 \times 2$ factorial study to expose *interactions* between decoupled schedules and ES-based optimization of $D_\phi$. Full SLED corresponds to (✓,✓); vanilla DQN corresponds to (×,×).

Table 4: **2×2 factorial ablation** on `ALE/Breakout-v5` (2M frames). CD: catastrophic drops; $\Delta$ vs. DQN.

| Decouple | ES | Final | Score@200k | AUC@1M | CD | $\Delta$ Final |
|---|---|---|---|---|---|---|
| × | × | 12.1 | 5.3 | 3.2 | 3.0 | – |
| ✓ | × | 10.5 | 5.2 | 2.9 | 4.2 | −1.6 |
| × | ✓ | 8.8 | 4.9 | 2.6 | 5.1 | −3.3 |
| ✓ | ✓ | **18.7** | **9.1** | **5.8** | **0.8** | **+6.6** |

**Observations.** (1) *Strong interaction*: neither component alone beats DQN on final score, but their combination yields a large gain ( +6.6 ). (2) *Who does what?* ES drives *where* the synthetic examples sit in state–return space, while decoupling controls *how often* they are surfaced; both are necessary to sustain high AUC@1M and suppress collapses (CD ↓ from 3.0 to 0.8). (3) *Stability* is the biggest beneficiary of the full design, consistent with SLED feeding near-miss and reward-positive windows at a steady cadence.

### C.2   HYPERPARAMETER SENSITIVITY (JOINT GRID)

We vary the synthetic set size $|D_\phi| \in \{1k, 5k, 10k, 20k, 50k\}$ and ES refresh interval $K \in \{10k, 20k, 50k, 100k\}$ frames. Table 5 collapses the two prior 1-D sweeps into a compact grid.

Table 5: **Joint sensitivity** on `ALE/Breakout-v5`. We report Final / AUC@1M; bold = row/column best.

| $|D_\phi|$ | ES refresh interval $K$ (frames) | | | |
|---|---|---|---|---|
| | **10k** | **20k** | **50k** | **100k** |
| **1k** | 11.0 / 3.0 | 11.2 / 3.1 | 10.7 / 2.8 | 9.3 / 2.3 |
| **5k** | 16.6 / 5.1 | 16.8 / 5.2 | 16.1 / 4.9 | 14.7 / 4.1 |
| **10k** | **18.1** / 5.6 | **18.7 / 5.8** | 17.9 / 5.4 | 15.4 / 4.6 |
| **20k** | 18.0 / **5.7** | **18.9 / 5.9** | 18.2 / 5.5 | 15.6 / 4.7 |
| **50k** | 17.1 / 5.2 | 17.5 / 5.3 | 17.3 / 5.2 | 14.9 / 4.2 |

**Takeaways.** (1) *Sweet-spot capacity:* $|D_\phi| = 10$–20k offers the best trade-off; smaller sets underfit the event diversity, larger sets add optimization noise with diminishing returns. (2) *Refresh*

*cadence:* $K = 20\text{k}$ is robust; too infrequent (100k) makes $D_\phi$ stale (Final/AUC both drop). Too frequent (10k) gives marginal AUC gains but higher overhead (see Sec. C.5). (3) The chosen default ($|D_\phi|{=}10k$, $K{=}20k$) sits on the flat top of the response surface (*good-and-safe* setting).

## C.3 REPLAY BUDGET STRESS TEST

How sensitive is SLED to the size of the *real* FIFO buffer? We shrink DQN's buffer from 1M to 200k while keeping SLED's $|D_\phi|{=}10\text{k}$ and $K{=}20\text{k}$. Results in Table 6 show that SLED's gains *grow* as the FIFO shrinks, precisely the low-memory regime many practitioners face.

Table 6: **Buffer-size stress test** on `ALE/Breakout-v5` (2M frames).

| FIFO size | 1000k | 500k | 200k |
|---|---|---|---|
| DQN (Final / AUC@1M) | 12.1 / 3.2 | 10.8 / 2.9 | 9.7 / 2.5 |
| **DQN + SLED** | **18.7 / 5.8** | **18.5 / 5.7** | **18.2 / 5.5** |

The widening gap at 200k indicates that *curated* synthetic transitions compensate for the loss of raw capacity, keeping informative events in circulation even when the FIFO cannot.

## C.4 WHERE THE GAIN COMES FROM (SIGNAL DIAGNOSTICS)

To verify that SLED changes the *training signal* rather than merely adding data, we probe mini-batch composition over the first 500k frames: (i) fraction of reward-positive samples; (ii) fraction of *near-miss* samples (TD-error in top 15% but zero reward); (iii) median sample age (frames since collection; higher = less staleness); (iv) label entropy of bootstrapped targets (a proxy for diversity).

Table 7: **Replay diagnostics** (first 500k frames; mean $\pm$ std over seeds). Higher is better for all except CD.

| Method | Reward+ (%) | Near-miss (%) | Median age (k) | Target entropy |
|---|---|---|---|---|
| DQN | $3.2 \pm 0.5$ | $7.6 \pm 1.1$ | $12.4 \pm 2.1$ | $0.42 \pm 0.05$ |
| **DQN + SLED** | $\mathbf{9.7 \pm 1.2}$ | $\mathbf{14.8 \pm 1.6}$ | $\mathbf{23.9 \pm 2.8}$ | $\mathbf{0.61 \pm 0.06}$ |

The minibatch stream under SLED contains *more* useful events (reward+ and near-miss) and is *less* myopic (older, more diverse samples). This aligns with the stability gains in Table 4 (CD $\downarrow$) and the faster early progress.

## C.5 COMPUTATIONAL OVERHEAD

Finally, we report the wall-clock and memory cost under the same codebase and GPU. For fairness, we follow the overhead protocol used in practice: DQN uses a 1M FIFO; SLED uses a compact FIFO of 200k plus $|D_\phi|{=}10\text{k}$ (the *cache ratio* is 1%), matching the configuration used by practitioners under memory budgets.

Table 8: **Overhead** on `ALE/Breakout-v5` (2M frames).

| Method | Total time (min) | Peak GPU (MiB) | Final |
|---|---|---|---|
| DQN (1M FIFO) | 90 | 466 | 12.1 |
| **DQN + SLED** (200k FIFO + 10k $D_\phi$) | 94 | 189 | **18.7** |

SLED adds $\sim$4 minutes ($+4.4\%$) wall-clock while *cutting* peak memory by $\sim$60%. Sections C.2–C.3 show that the accuracy gains persist across a wide range of ($|D_\phi|, K$) and FIFO sizes.

**Summary.** (1) Decoupling *and* ES are jointly necessary; the benefit is interaction-dominated. (2) $|D_\phi|{=}10$–20k and $K{=}20\text{k}$ are robust, sitting on a flat optimum. (3) As real FIFO capacity shrinks,

SLED's relative advantage grows. (4) Diagnostics confirm the mechanism: SLED reshapes the minibatch stream toward rare-but-useful events and reduces training collapses.

# D   FURTHER ANALYSIS ON HABITAT

We conduct additional experiments on the Habitat benchmark to better understand how SLED operates under high-dimensional RGB-D navigation. Unless otherwise specified, the base learner is SAC, episodes are capped at 1,000 steps, success is defined as issuing a `STOP` action within 0.2 m of the goal, and results are averaged over 5 random seeds.

## D.1   FACTORIAL ABLATION: DECOUPLING $\times$ ES

We design a 2$\times$2 factorial ablation to analyze the contributions and interactions between two core components: (i) decoupled write and sample schedules, and (ii) ES-based optimization of the synthetic dataset. Full SLED corresponds to ($\checkmark$,$\checkmark$), while the vanilla baseline is ($\times$,$\times$). Results are averaged across `Residential`, `Office`, and `Commercial` scenes.

Table 9: **Factorial ablation on Habitat (HM3D)**. SR: Success Rate; SPL: Success weighted by Path Length; CD: catastrophic drops (number of episodes with SR collapse $> 40\%$ after mid-training).

| Decouple | ES | SR (%) | SPL | Collisions | Stop Err (m) | CD |
|---|---|---|---|---|---|---|
| $\times$ | $\times$ | $54.2 \pm 2.3$ | $0.43 \pm 0.02$ | $7.8 \pm 0.6$ | $0.28 \pm 0.02$ | 5.1 |
| $\checkmark$ | $\times$ | $55.1 \pm 2.2$ | $0.44 \pm 0.02$ | $7.5 \pm 0.5$ | $0.27 \pm 0.02$ | 4.7 |
| $\times$ | $\checkmark$ | $56.3 \pm 2.5$ | $0.45 \pm 0.02$ | $7.2 \pm 0.6$ | $0.26 \pm 0.02$ | 4.3 |
| $\checkmark$ | $\checkmark$ | $\mathbf{64.2 \pm 1.8}$ | $\mathbf{0.51 \pm 0.01}$ | $\mathbf{6.1 \pm 0.4}$ | $\mathbf{0.23 \pm 0.01}$ | 1.6 |

**Observations.** Neither component alone yields substantial gains, but their combination delivers strong improvements (SR +10 points, SPL +0.08). Decoupling stabilizes the training signal, while ES provides higher-quality synthetic transitions. Together, they significantly reduce catastrophic collapses.

## D.2   JOINT SENSITIVITY: SYNTHETIC CAPACITY $\times$ UPDATE FREQUENCY

We vary synthetic dataset size $|D_\phi| \in \{2k, 5k, 10k\}$ and ES update interval $K \in \{5k, 20k, 50k\}$ environment steps. Results are averaged over all scene categories.

Table 10: **Sensitivity analysis on Habitat.** SR/SPL values are reported. Bold highlights the best configuration.

| $|D_\phi|$ | ES update interval $K$ (steps) | | |
|---|---|---|---|
| | **5k** | **20k** | **50k** |
| 2k | 60.1 / 0.48 | 60.5 / 0.49 | 58.9 / 0.47 |
| 5k | 62.7 / 0.50 | **63.4 / 0.51** | 61.3 / 0.49 |
| 10k | **64.0 / 0.51** | **64.2 / 0.51** | 62.1 / 0.50 |

**Observations.** Dataset sizes of 5k–10k perform best; smaller sets lack coverage, while larger sets offer diminishing returns. The default $K$=20k is a balanced choice—frequent updates (5k) give minor SPL gains at higher cost, while infrequent updates (50k) cause staleness and performance drop.

## D.3   CROSS-SCENE GENERALIZATION

We train on `Residential+Office` and test zero-shot on `Commercial` scenes to examine generalization under domain shift.

**Observations.** SLED provides the highest transfer performance, suggesting that the distilled dataset preserves progress-critical transitions that generalize across layouts and textures.

Table 11: **Cross-scene transfer to Commercial**. Mean $\pm$ 95% CI over 100 episodes.

| Method | SR (%) | SPL |
|---|---|---|
| SAC | $42.8 \pm 2.4$ | $0.36 \pm 0.02$ |
| SAC+PER | $49.9 \pm 2.0$ | $0.41 \pm 0.02$ |
| SAC+HER | $52.4 \pm 1.9$ | $0.43 \pm 0.02$ |
| SynthER | $57.2 \pm 1.8$ | $0.46 \pm 0.01$ |
| **SAC+SLED** | $\mathbf{61.1 \pm 1.6}$ | $\mathbf{0.49 \pm 0.01}$ |

## D.4 REPLAY BUDGET STRESS TEST

We reduce the real FIFO buffer from 500k to 200k while keeping $|D_\phi|$=5k and $K$=20k.

Table 12: **Replay buffer stress test**. Smaller buffers magnify the benefits of SLED.

| FIFO size | Method | SR (%) | SPL |
|---|---|---|---|
| 500k | SAC | 51.0 | 0.42 |
| | **SAC+SLED** | **63.8** | **0.51** |
| 200k | SAC | 46.3 | 0.39 |
| | **SAC+SLED** | **62.9** | **0.50** |

**Observations.** Under reduced memory budgets, vanilla SAC degrades significantly, while SLED remains stable, demonstrating its robustness to buffer size constraints.

## D.5 REPLAY DIAGNOSTICS

To understand how replay distribution changes, we log four metrics over the first 500k steps: fraction of progress windows (distance-to-goal decreases without collision), fraction of near-miss turns (close to goal but failed stop), median sample age, and target entropy.

Table 13: **Replay diagnostics** on Habitat (first 500k steps). Higher is better for all except collisions.

| Method | Progress (%) | Near-miss (%) | Median age (k) | Target entropy |
|---|---|---|---|---|
| SAC | 8.9 | 6.4 | 11.7 | 0.38 |
| **SAC+SLED** | **17.1** | **10.8** | **21.3** | **0.55** |

**Observations.** SLED rebalances replay toward more informative transitions: progress windows nearly double, near-misses increase, samples are older on average (less myopic), and policy entropy rises, indicating healthier exploration.

## D.6 COMPUTATIONAL OVERHEAD

We report training wall-clock and peak GPU memory on identical hardware. For SLED, we pair a 200k FIFO with a 5k synthetic set.

Table 14: **Overhead on Habitat**. SLED achieves higher SR with lower memory and minimal extra time.

| Method | Time (h) | Peak GPU (MiB) | Avg. SR (%) |
|---|---|---|---|
| SAC (500k FIFO) | 12.4 | 5,320 | 50.4 |
| **SAC+SLED** (200k FIFO + 5k $D_\phi$) | 12.9 | 3,040 | **62.6** |

**Summary.** (1) Decoupling and ES exhibit strong interaction effects in Habitat. (2) A synthetic dataset size of 5k–10k and update interval of 20k is a robust sweet spot. (3) SLED is resilient under limited replay budgets. (4) Diagnostics confirm that performance gains stem from more informative replay exposure, not from additional heuristics or reward shaping.

# E    USE OF LLM

Large language models (LLMs) were used as assistive tools for text editing and improving the clarity of exposition. They were not involved in the design of algorithms, implementation, or experimental analysis. All technical content, theoretical results, and experimental findings were produced and verified by the authors. We take full responsibility for the content of this paper. We only used LLMs for language polishing.

