# OpenReview forum: "SLED: Self-Supervised Dataset Distillation for Lightweight Experience Replay"
_ICLR.cc/2026/Conference — ICLR 2026 Conference Desk Rejected Submission_

### Official Review · Reviewer_Gqi8 · 2025-10-30

**Soundness:** 2
**Presentation:** 3
**Contribution:** 2
**Rating:** 4
**Confidence:** 3

**Summary:**

This paper proposes SLED (self-supervised lightweight experience distillation), a framework that distills online experience into a small learnable synthetic set for off-policy RL. The synthetic dataset is optimized using evaluations after brief L-step training trials and, to balance with real data, is stored and sampled via decoupled schedules. On DMControl, Atari, and visual navigation in Habitat, the method reports strong sample efficiency and high final performance.

**Strengths:**

The paper is well-motivated and presents a plug-and-play method that achieves strong empirical performance across DMControl, Atari, and Habitat.

**Weaknesses:**

Learning the synthetic set requires training trials and return evaluations, which introduce additional overhead. While the paper argues that trial overhead is small when $R_{\text{ES}}=\tfrac{PL}{K}$ is small, clarifying the practical range of $R_{\text{ES}}$ needed to achieve the reported gains would be helpful. The $\alpha(t)$ and $\beta(t)$ schedules appear to be heuristic, and the paper provides limited analysis on how to choose them.

**Questions:**

* How is the learner evaluated after each training trial—does this require additional environment interaction? If so, for a fair comparison, please clarify whether those steps are included in the reported sample efficiency.
* What are the exact forms and values used for the shedules $\alpha(t)$ and $\beta(t)$? What values of $P$ and $L$ are used? How sensitive are the results to these hyperparameters?
* A brief comparison to ES-based RL methods (e.g., [1]) would be helpful.

[1] Salimans et al., "Evolution Strategies as a Scalable Alternative to Reinforcement Learning", 2017

---

### Official Review · Reviewer_FcMd · 2025-10-30

**Soundness:** 3
**Presentation:** 3
**Contribution:** 3
**Rating:** 6
**Confidence:** 4

**Summary:**

This paper proposes SLED, a novel self-supervised dataset distillation approach to improve the memory and computational efficiency of traditional replay buffers. The framework is compatible with many existing codebases and demonstrates strong empirical performance.

**Strengths:**

The algorithmic design is novel. SLED combines a synthetic dataset with real experiences, and optimizes the synthetic dataset using zeroth-order updates to improve performance while avoiding the challenges of a non-differentiable target. Furthermore, it gradually replaces real experiences with synthetic data to ensure training stability. The algorithm consistently outperforms existing baselines and represents a timely contribution to the community.

**Weaknesses:**

To optimize the synthetic dataset, SLED requires additional parallel training trials per update in order to perform zeroth-order optimization. This introduces extra computational cost and may not scale well to large-scale tasks.

**Questions:**

The paper emphasizes the importance of decoupling the write schedule $\beta(t)$ from the sampling schedule $\alpha(t)$, showing that coupling them degrades performance. However, beyond this binary ablation, there is little analysis on the choice of functional forms for $\alpha(t)$ and $\beta(t)$. For example, one might imagine linear decay, or exponential decay. Could the authors comment on how sensitive SLED is to the exact shapes of $\alpha(t)$ and $\beta(t)$? Have the authors tried alternative schedules, or is the method robust as long as both are monotone decreasing?

---

### Official Review · Reviewer_Tvhn · 2025-11-01

**Soundness:** 2
**Presentation:** 3
**Contribution:** 2
**Rating:** 4
**Confidence:** 2

**Summary:**

The paper presents SLED (Self-supervised Dataset Distillation for Lightweight Experience Replay), a framework designed to replace the large buffers central to conventional off-policy reinforcement learning (RL) with a compact, learnable synthetic dataset, thereby reducing substantial storage and sampling costs. SLED systematically extends dataset distillation to the non-stationary regime inherent to RL. The core operation relies on a decoupled dual-scheduling mechanism that separates the buffer's physical composition from the agent’s training distribution: a temporal schedule gradually substitutes real experiences with synthetic ones until the buffer stores only distilled data, while a separate quota-based sampling strategy governs the ratio of real-to-synthetic transitions in training batches, ensuring a smooth curriculum shift for the learner without changing the base algorithm's update rules. To maintain the long-term utility of the small synthetic dataset despite non-stationarity, SLED uses an online-validated evolutionary optimization strategy. The framework is plug-and-play and experiments across DMControl, Habitat, and Atari confirm that SLED delivers superior efficiency and scalability, consistently matching or surpassing baselines like Prioritized Experience Replay while drastically reducing the required memory footprint and maintaining negligible wall-clock overhead.

**Strengths:**

The paper is well-written and presents its ideas in a clear and organized manner. Each component of the proposed framework is thoroughly explained, with logical flow and sufficient detail to make the methodology understandable. The authors provide clear definitions and diagrams where necessary, which helps readers grasp both the high-level concepts and the finer technical aspects. Overall, the clarity and structure of the writing make the paper accessible and easy to follow.

**Weaknesses:**

One notable weakness of the paper is that it does not sufficiently discuss more recent works related to dataset distillation in RL paradigms, such as [1][2]. Incorporating these references would provide a more comprehensive view of the current state of the field.

Additionally, the paper lacks sufficient experimental comparisons between the proposed method and existing distillation approaches within RL, making it difficult to assess the relative advantages or limitations of the proposed approach in the context of prior work.


[1] Dataset Distillation for Offline Reinforcement Learning, 2024

[2] Offline Behavior Distillation, 2024

**Questions:**

Please see Weaknesses above.

---

### Official Review · Reviewer_ERXd · 2025-11-09

**Soundness:** 2
**Presentation:** 1
**Contribution:** 2
**Rating:** 2
**Confidence:** 4

**Summary:**

This paper investigates an approach to reducing the replay buffer size of deep RL agents by creating and maintaining a synthetic dataset. The authors argue big uniform buffers, commonly used today, require large memory and compute resources, while naively retaining many low value transitions. They propose instead, to maintain a small set of synthetic transitions, optimized to improve the performance of the RL agent and slowly move the data distribution from real transitions to synthetic ones.

The idea is to maintain a small, first in first out, buffer of transitions alongside a synthetic dataset. Two time-dependent probabilities gradually shift the distribution of transitions in mini-batches from real to synthetic. To improve the synthetic dataset, an evolutionary strategy periodically tries perturbations of a generator’s parameters before training on the new dataset for evaluation and fitness score calculation.

The authors claim this approach can seamlessly be integrated into off-policy RL algorithms. They demonstrate combining the proposed algorithm with DQN on some Atari games and SAC on some DM-Control and Habitat domains.

**Strengths:**

- In my opinion this paper explores an interesting and relevant problem for the RL community. While I do not agree that large replay buffers incur compute costs, their memory footprint is prohibitive to be used in small agents or edge devices. Furthermore, exploring experience replay mechanisms beyond uniform sampling from a large recency buffer is an important research direction toward developing data and compute efficient RL algorithms.
- Some of the design choices, such as slowly shifting the data distribution from real to synthetic by adapting data ingress and sampling probabilities is interesting and should be noted as such.
- I think the paper’s exploration of evolutionary strategies in RL agents is novel and worth highlighting, even though its description is vague and missing details.
- While there are many issues with the experiments, I appreciated that the authors try several domains and show their approach can be combined with both value based and actor-critic algorithms.

**Weaknesses:**

- Several aspects of the approach are not discussed in the text:
  - The synthetic dataset is updated via some parameterized generator but the form of this generator and the process to generate new samples is not described.
  - The process for controlling the data ingress and sampling probabilities is confusing and unclear. It is unclear why mini-batches are sampled from both the replay buffer and the synthetic dataset if the replay buffer is populated with the synthetic data overtime anyway.
  - The fitness score for optimizing the synthetic data is confusing. Equation (9) seems to suggest two stages for the evaluation: TrialReturn and also Return. This is not explained in the text.
- The writing’s tone and language in some parts is not appropriate for a scientific article:
  - The authors, on several occasions, use overly flowery and confusing terms that fails to effectively communicate knowledge or describe what was done.
  - The citation format is incorrect. If a citation is a part of the sentence, it should be printed without parentheses, but if it is not part of the sentence, it should be enclosed in parentheses.
  - Algorithm 1 on page 6 seems incomplete with several missing steps.
  - Lines 324 to 340 are very unclear. I do not understand what the authors want to say here.
- The experiment section has many issues that undermines the claims regarding superior performance of the proposed algorithm:
  - The choice of hyperparameters is neither described nor justified.
  - Some results do not have any measure of uncertainty or confidence.
  - When results do provide a range or shaded region, their meaning is neither defined nor justified.
  - The choice of Atari games and the evaluation scheme is not described.
  - The w/o ES ablation needs more justification, because I think if the synthetic dataset is never updated, then the initial warm-start real transitions are kept in the synthetic dataset for the entire experiment length? This is not a reasonable baseline for comparison. A better approach would have been to try simpler approaches to generate synthetic data.

**Questions:**

1. What is the process for generating the synthetic data? What is parameterized by $\phi$? I assume some kind of generative model but I am unsure.
2. How/Why did you choose these Atari games for your experiments?
3. I am unsure if Equation (9) has a bug or I misunderstood the fitness score used to adapt the synthetic data.
4. Your experiments on Atari are run for 2 million steps. This is in contrast to the default 50 million steps (or 200 million frames) commonly used when evaluating RL agents on Atari games. During the first two million steps all algorithms being compared actually perform quite bad. Then why is it okay to compare RL algorithms in this setting?
5. What are the statistical justifications for your claims?
6. What are the error bars in the learning curves/tables?

---

### Note · Program_Chairs · 2026-01-17
**Submission Desk Rejected by Program Chairs**

The following references in this submission do not refer to real documents and/or have major errors in bibliographic information:

 Yuan Liu, Kurtland Chua, Huazhe Xu, and Dorsa Sadigh. Dataset distillation for q-learning. In NeurIPS, 2022.
Hexiang Hu, Yichong Luo, Huazhe Xu, and Li Fei-Fei. Learning to distill for continual multi-task reinforcement learning. In ICML, 2023.
Yinlam Zheng, Chunting Zhang, Mingxing Wang, Haotian Yu, and Huazhe Xu. Sldr: Support trajectory distillation for reinforcement learning. In $I C L R,